# The pharmacodynamic and differential gene expression analysis of PPAR α/δ agonist GFT505 in CDAHFD-induced NASH model

Linfu Liu[1,2☯], Chuang Liu[1,2☯], Manyu Zhao[1], Qianru Zhang[2,3], Ying Lu[2], Ping Liu[4], Hua Yang[1], Jinliang Yang[2,4], Xiaoxin Chen[3,4]*, Yuqin Yao[1,2]*

**1** West China School of Public Health and West China Fourth Hospital, Sichuan University, Chengdu, China, **2** Collaborative Innovation Center for Biotherapy and Cancer Center, West China Hospital, Sichuan University, Chengdu, China, **3** Guangdong Zhongsheng Pharmaceutical Co., Ltd., Guangdong, China, **4** Department of Gynecology and Obstetrics, West China Second Hospital, Sichuan University, Chengdu, Sichuan, China

☯ These authors contributed equally to this work.
* yuqin_yao@scu.edu.cn (YY); chenzhenyu2000@zspcl.com (XC)

**Data Availability Statement:** The data has been uploaded to NCBI Sequence Read Archive and the relevant accession number is PRJNA657593.

## Abstract

Peroxisome proliferator-activated receptor α/δ (PPAR α/δ), regulating glucolipid metabolism and immune inflammation, has been identified as an effective therapeutic target in non-alcoholic steatohepatitis (NASH). Dual PPAR α/δ agonist, such as GFT505 (also known as elafibranor), demonstrated potential therapeutic effect for NASH in clinical trials. To profile the regulatory network of PPAR α/δ agonist in NASH, the choline-deficient, L-amino acid-defined, high-fat diet (CDAHFD) induced NASH model was used to test the pharmacodynamics and transcriptome regulation of GFT505 in this study. The results showed that GFT505 ameliorated hepatic steatosis, inflammation and fibrosis in CDAHFD mice model. RNA-sequencing yielded 3995 up-regulated and 3576 down-regulated genes with GFT505 treatment. And the most significant differentialy expressed genes involved in glucolipid metabolism (*Pparα*, *Acox1*, *Cpt1b*, *Fabp4*, *Ehhadh*, *Fabp3*), inflammation (*Ccl6*, *Ccl9*, *Cxcl14*) and fibrosis (*Timp1*, *Lamc3*, *Timp2*, *Col3a1*, *Col1a2*, *Col1a1*, *Hapln4*, *Timp3*, *Pik3r5*, *Pdgfα*, *Pdgfβ*, *Tgfβ1*, *Tgfβ2*) were confirmed by RT-qPCR. The down-regulated genes were enriched in cytokine-cytokine receptor interaction pathway and ECM-receptor interaction pathway, while the up-regulated genes were enriched in PPAR signaling pathway and fatty acid degradation pathway. This study provides clues and basis for further understanding on the mechanism of PPAR α/δ agonist on NASH.

## Introduction

Non-alcoholic fatty liver disease (NAFLD) is a chronic liver disease usually accompanied by obesity, type 2 diabetes, metabolic syndrome and cardiovascular diseases [1, 2]. NASH is a progressive disease from NAFLD, which characterized by steatosis, hepatocellular ballooning, lobular inflammation and fibrosis. Without any effective intervention, NASH could develop into cirrhosis and hepatocellular carcinoma [3].

**Funding:** This work was supported by the National Major Scientific and Technological Special Project (Nos. 2018ZX09201002, 2018ZX09711001-011 and 2019ZX09201001) and the National Science Foundation of China (No. 81773375). The authors thank the research platform provided by Public Health and Preventive Medicine Provincial Experiment Teaching Center at Sichuan University.

**Competing interests:** All the authors declare that they have no conflict of interests. And all the institutions and companies declare that they have no conflict of interests. In this study, Sichuan University, including Collaborative Innovation Center for Biotherapy and Cancer Center, West China School of Public Health and West China Fourth Hospital, provided platforms for performing the experiments, but did not have any additional role in the study design, data collection and analysis, decision to publish, or preparation of the manuscript. Guangdong Zhongsheng Pharmaceutical Co., Ltd employs [Qianru Zhang and Xiaoxin Chen] but does not fund the study and alter our adherence to PLOS ONE policies on sharing data and materials. The specific roles of the authors are articulated in the 'author contributions' section.

Lipid toxicity results in the liver injury and hepatitis [4]. Persistent inflammatory injury over activates injury repair process, thus leading to hepatic fibrosis [5]. Nuclear receptors PPARs are ligand activated transcription factors regulating lipid metabolism and inflammation. PPAR α, which is mainly expressed in liver, kidney, heart and skeletal muscle, modulates lipid metabolism by regulating fatty acid transportation and β-oxidation [6]. Activating of PPAR α could decrease the lipid accumulation and oxidative stress in liver [7]. Similar to PPAR α, PPAR δ is also involved in lipoprotein metabolism and plays an important role in the anti-inflammatory effect. Study has showed that the PPAR δ agonist could improve hepatic steatosis in NASH [8]. GFT505, a dual PPAR α/δ agonist, has been reported to be effective in clinical trials of NASH [9]. A short-term (4–8 weeks) phase II study showed that GFT505 reduced the plasma triglyceride level, increased high-density lipoprotein-cholesterol (HDL-c) and reduced low-density lipoprotein- cholesterol (LDL-c) in prediabetic patients [10]. Other studies in abdominal obese patient reported that GFT505 improved peripheral insulin sensitivity and significantly reduced the liver enzyme concentrations of AST and ALT [11, 12]. These clinical studies demonstrated that GFT505 is a potential drug for NASH treatment by regulating lipid and glucose metabolism.

In this study, in order to further understand the mechanism of the PPAR α/δ agonist on NASH, the pharmacodynamics and transcriptome regulation of GFT505 were evaluated in CDAHFD fed mouse NASH model. The effects of GFT505 on fatty acid metabolism, inflammation and fibrosis were determined by histological and immunohistochemical analysis. The differential gene expression regulated by GFT505 in NASH was analyzed by RNA-sequencing. Bioinformatics analysis was used to investigate the effects of GFT505 on biological processes and biological pathways. The representative different expressed genes (DEGs) were validated by RT-qPCR.

## Materials and methods

### Animals, drug and cell line

C57BL/6J mice (male, 4-week-old) were obtained from the Vital-River Animal Ltd (Beijing, China). All experimental procedures were approved by the institutional animal care and treatment committee of West China Hospital in Sichuan University (approval no. 2019273A), according to the National Institutes of Health guide for the care and use of laboratory animals. Animals were housed in a specific pathogen-free environment and maintained on standard diet for 4 weeks at a temperature of $24 \pm 1^\circ C$ and 12/12 h light/dark cycles, with free access to food and water. Mice at 9 weeks of age were fed with normal diet or the CDAHFD diet (60 kcal% fat, Research Diets Inc, New Brunswick, New Jersey, USA, A06071302).

Normal human hepatic cell line LO2 was obtained from the Chinese Academy of Sciences (Shanghai, China). The cells were cultured in RPMI-1640 medium (Gibco, CAT# C11875500BT) supplemented with 10% fetal bovine serum (ZETA, CAT# Z7185FBS-500), 100 U/mL penicillin/streptomycin (Gibco, CAT# 15140–122). GFT505 (Med Chem Express, Shanghai, China) were dissolved in the solvents (1% carboxymethyl cellulose with 0.1% tween 80) for animal experiments and in DMSO for cell experiments.

### CDAHFD-fed mouse NASH model

Mice were divided into five groups (10 mice in each group): Control group (mice fed on normal diet), vehicle group (mice fed on CDAHFD diet treated with vehicle), and GFT505 treatment groups (mice fed on CDAHFD diet and treated with 3 mpk, 10 mpk, 30 mpk GFT505). The GFT505 administration started at the fifth week and ended at the twelfth week. The blood glucose was analyzed once a week before the end of the experiment, terminal blood sample

was collected from the heart in non-fasted mice and used for serum biochemical analysis. Animals were anesthetized with 3% pentobarbital sodium by intraperitoneal injection. The terminal blood samples were obtained from the eyelids for biochemical analysis at the twelfth week and the tissues were collected for histological and immunohistochemical analysis. Liver tissues for RNA-seq were stored in liquid nitrogen before RNA extraction. Serum chollesterol levels detected by Cholesterol Gen.2 (CHOL2) kit (Roche, CAT# 03039773190). Serum TG levels detected by Triglycerides (TRIGL) kit (Roche, CAT# 06380115190). Serum AST levels detected by Aspartate Aminotransferase (ASTL) kit(Roche, CAT# 06380115190). Serum ALT levels detected by Alanine aminotransferase acc. to IFCC (ALTL) kit (Roche, CAT# 04718569190).

## Histological and immunohistochemical assays

Liver tissues were fixed in 10% neutral formalin, embedded in paraffin, and cut into 5 μm sections. Hematoxylin and eosin (H&E) staining was used to evaluate liver steatosis, inflammation. Hepatocellular steatosis, ballooning and inflammation of the specimens were scored according to the standards as described before [13]. Hepatic fibrosis were evaluated by Sirius red staining and analyzed by Nano Zoomer Digital Pathology S210 and Image-Pro Plus 6.0. The Sirius Red staining positive areas were presented as the percentage of the total area of the specimen.

For immunohistochemistry, liver sections were incubated with primary monoclonal antibodies against α-smooth muscle actin (Abcam, CAT# ab3471) and collagen I (Huaan Biotechnology, CAT# ET1607-53). The expression of α-SMA and collagen I in the liver tissue were measured and quantified by Image-Pro Plus 6.0.

## Sample preparation, library construction and RNA-seq

The total RNA extraction, library construction and sequencing of liver tissues (n = 3) were performed at the Beijing Novogene Corporation. Tiangen RNA prep Pure Plant Kit (Tiangen, CAT# DP441) was used to extract total liver RNA from the three mice of each group. The extracted RNA quantity and quality were verified by Qubit® RNA Assay Kit (Life Technologies, CAT# Q32852) in Qubit® 2.0 Fluorometer (Life Technologies, CA, USA) and the RNA Nano 6000 Assay Kit (Agilent Technologies, CAT# 5067–1511) of the Bioanalyzer 2100 system (Agilent Technologies, CA, USA). The cDNA library construction and RNA sequencing were performed as described before [13].

## Differential gene expression and bioinformatics analysis

Clean data were obtained from raw data by RNA-seq after removing reads containing adapter, ploy-N and with low quality. DEGs analysis was based on clean data. Bowtie v2.0.6 was used to build the indexes of the reference genome and TopHat v2.0.9 was used to align the paired-end clean reads and reference genome. HTSeq v0.6.1 was used to count the read numbers mapped of each gene. DESeq R package (1.10.1) was used to analyze the differential expressed genes between two groups. The $p$-values of each result were adjusted by the Benjamini and Hochberg's approach to control the false discovery rate. The gene with an adjusted $P$-value $< 0.05$ was identified as DEG by DESeq. GO seq R package were used for the Gene Ontology (GO) analysis. $p < 0.05$ was identified as the significant enriched GO terms. KOBAS software was applied to the Kyoto encyclopedia of genes and genomes (KEGG) pathway enrichment analysis.

## Quantitative real-time polymerase chain reaction assay (RT-qPCR)

Total RNA of liver tissues of mice (n = 3) in each group were extracted using TRIzol (Invitrogen, CAT# R1100). RNA degradation and contamination were monitored on 1% agarose gels.

RNA purity was checked using the Nano Photometer® spectrophotometer (IMPLEN, CA, USA). RNA concentration was measured using Qubit® RNA Assay Kit (Life Technologies, CAT# Q32852) in Qubit® 2.0 Flurometer (Life Technologies, CA, USA). RNA integrity was assessed using the RNA Nano 6000 Assay Kit (Agilent Technologies, CAT# 5067–1511) of the Bioanalyzer 2100 system (Agilent Technologies, CA, USA). First-strand cDNA was synthesized using HiScriptII Q RT Supermix for qPCR (Vazyme, CAT# R223-01). Quantitative RT-PCR was carried out using Aceq QPCR SYBR Green Master Mix (Vazyme, CAT# Q111-02/03). Both of them were carried out according to the instructions of manufacturer. The corresponding primer sequences were listed in S5 Table.

## Lipid accumulation cell model

Human LO2 cell line were seeded into 6-well plates and maintained in RPMI 1640 (Gibco, CAT# C11875500BT) medium supplemented with 10% fetal bovine serum (ZETA, CAT# Z7185FBS-500) and 100 U/mL penicillin/streptomycin (Gibco, CAT# 15140–122) for 24 h. The LO2 cells were treated with oleic acid/palmitate mixture (2:1 M ratio, final concentration 0.8 mM) for 24 h. And then cells were exposed to 7.5 or 15 μM GFT505. TG content (Nanjing jiancheng Biological Engineering institute, CAT# A110-1) and Oil red O staining (Sigma, CAT# O1391) were performed to evaluate the lipid accumulation according to the manufacturer´s instruction.

## Statistics analysis

The data of all experiment results were presented as mean ± SEM. The difference among groups was analyzed by Student's t-test or one-way ANOVA followed by LSD multiple comparison analysis (GraphPad prism 6.0, La Jolla, CA). $p < 0.05$ was considered to be significant.

# Results

## GFT505 improved the liver metabolism in the CDAHFD-induced NASH mice model

As shown in Fig 1A and 1B, although there was no difference in body weight between the GFT505 treatment groups and vehicle group, the ratio of liver weight to body weight kept increasing in a dose-dependent manner. And treatment with GFT505 also increased the concentration of serum cholesterol (Fig 1C), but had no effect on serum TG expression (Fig 1D). Importantly, the concentrations of AST were decreased at the dosages of 10 and 30 mpk of GFT505 (Fig 1E) and ALT were significantly reduced after treated with all the dosages of GFT505 (3, 10 and 30 mpk) (Fig 1F).

## GFT505 attenuated the liver steatosis, inflammation and fibrosis in NASH

To determine the effects of GFT505 on liver steatosis, inflammation and fibrosis in NASH, hematoxylin and eosin (H&E) staining and Sirius red staining were applied. The results of H&E staining demonstrated that GFT505 inhibited the steatosis and inflammation of NASH in a dose-dependent manner (Fig 2A). Sirius red staining showed that GFT505 attenuated the fibrosis in NASH mice model induced by CDAHFD (Fig 2B). Pathological score of H&E staining results were presented in Fig 2C and 2D. GFT505 of 10 and 30 mpk attenuated liver steatosis by 46% and 53%, respectively (Fig 2C). And GFT505 at the doses of 3, 10 and 30 mpk suppressed the CDAHFD-induced inflammation by 33%, 44% and 50% respectively (Fig 2D). Furthermore, the pathological scores of Sirius red staining showed that GFT505 suppressed fibrosis by 65% (10 mpk) and 58% (30 mpk) (Fig 2E). RT-qPCR results showed that the CD45

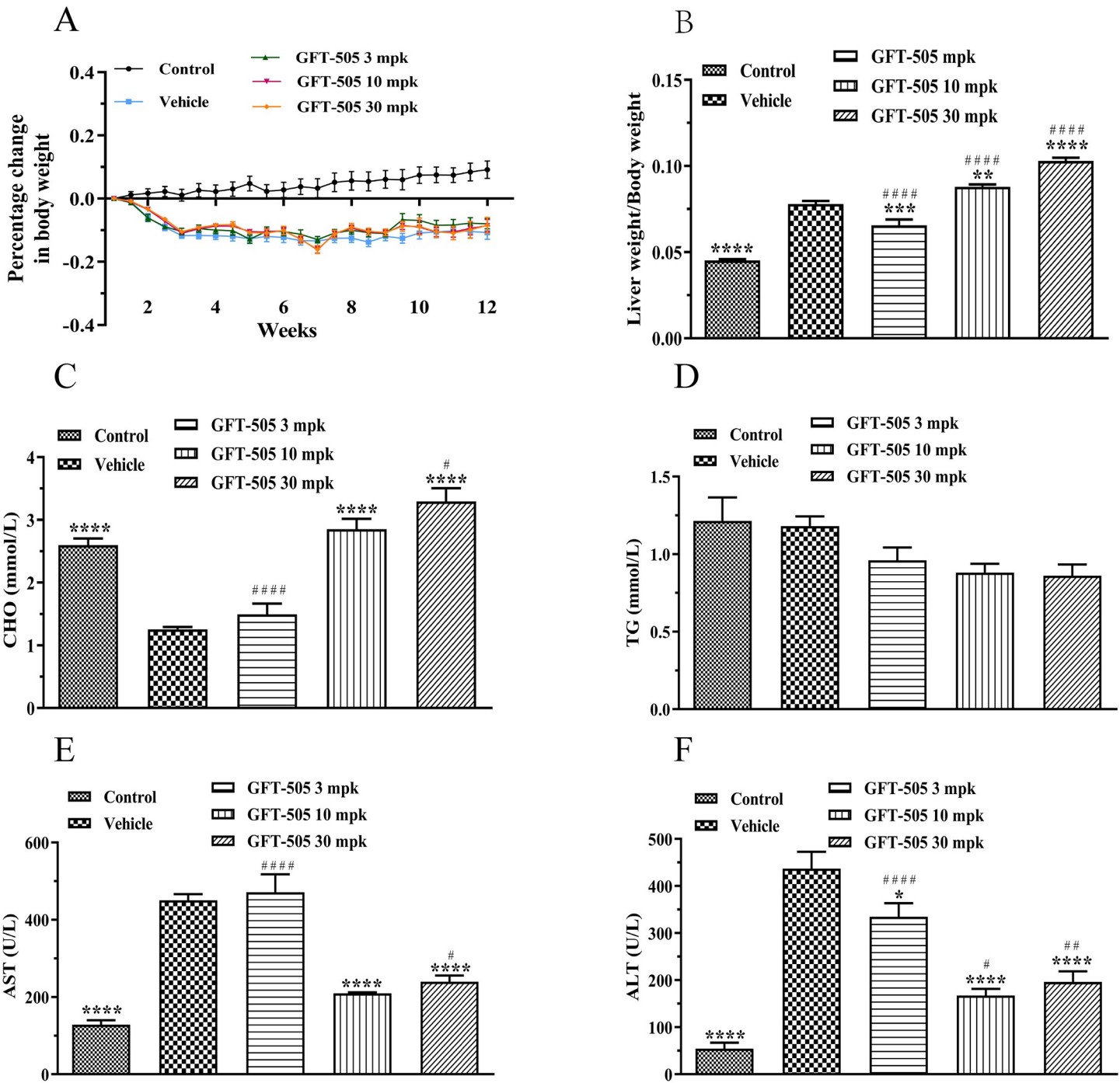

**Fig 1. Metabolic changes observed following long-term treatment with GFT505 in C57BL/6 mice.** (A) Percentage change in body weight for 12 weeks. (B) Ratio of liver weight to body weight. (C) Serum CHO levels detected by Cholesterol Gen.2 (CHOL2) kit. (D) Serum TG levels detected by Triglycerides (TRIGL) kit. (E) Serum AST levels detected by Aspartate Aminotransferase (ASTL) kit. (F) Serum ALT levels detected by Alanine aminotransferase acc. to IFCC (ALTL) kit. Control, normal diet; Vehicle and GFT505 groups, CDAHFD diet. Statistic analysis were performed using one-way ANOVA; Values were means ± SEM; $^{*}p < 0.05$, $^{**}p < 0.001$, comparing with vehicle group; $^{#}p < 0.05$, $^{##}p < 0.001$, comparing with control group; n = 10.

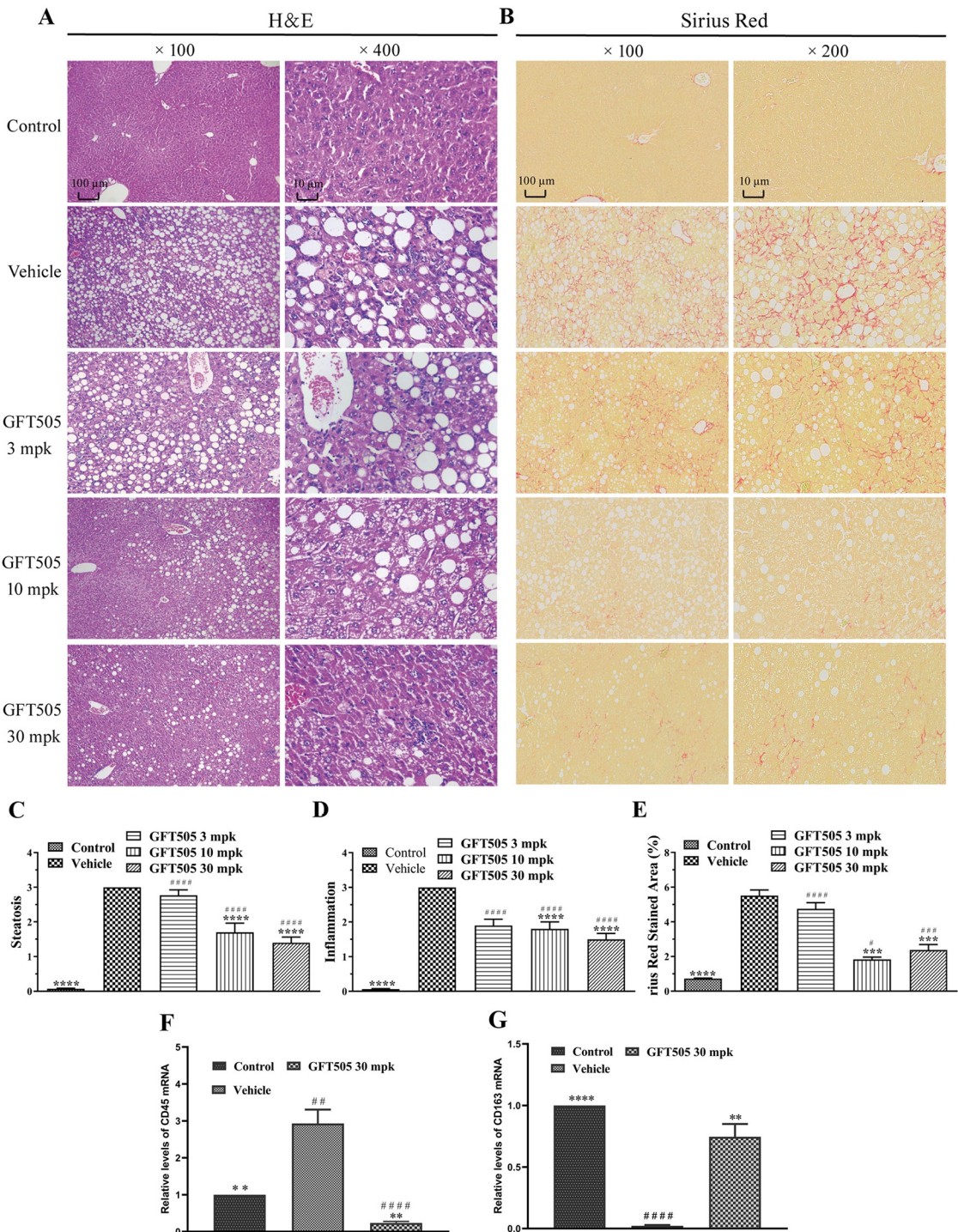

**Fig 2. GFT505 attenuated hepatic steatosis, inflammation and fibrosis in CDAHFD-fed C57BL/6 mice.** (A) Results of liver Hematoxylin and Eosin (H&E) staining. (B) Results of liver Sirius Red staining. (C, D) NASH/NAFLD Clinical Research Network scoring system and the scores in steatosis and inflammation. (E) Sirius red stained area quantification. (F) Relative mRNA expression of the CD45 by RT-qPCR. (G) Relative mRNA expression of the CD163 by RT-qPCR; Control, normal diet; Vehicle and GFT505 groups, CDAHFD diet; Statistic analysis were performed using one-way ANOVA; Values were means ± SEM; $^*p < 0.05$, $^{**}p < 0.001$, compared with vehicle group; $^\#p < 0.05$, $^{\#\#}p < 0.001$, comparing with control group; n = 10.

(M1-macrophage marker) was higher in the vehicle group compared with the control group and the GFT505 (30 mpk) group (Fig 2F). The CD163 (M2-macrophage marker) was lower in the vehicle group compared with the control group and the GFT505 (30 mpk) group (Fig 2G). The effects of GFT505 on liver steatosis and fibrosis were further confirmed by Oil Red O staining and immunohistochemistry. Treating with 30 mpk GFT505 reduced lipid droplets accumulation (Fig 3A) and the quantification was showed in Fig 3C. Decreased protein concentrations of α-SMA (Fig 3B and 3D) and collagen I (Fig 3B and 3E) were demonstrated after GFT505 treatment.

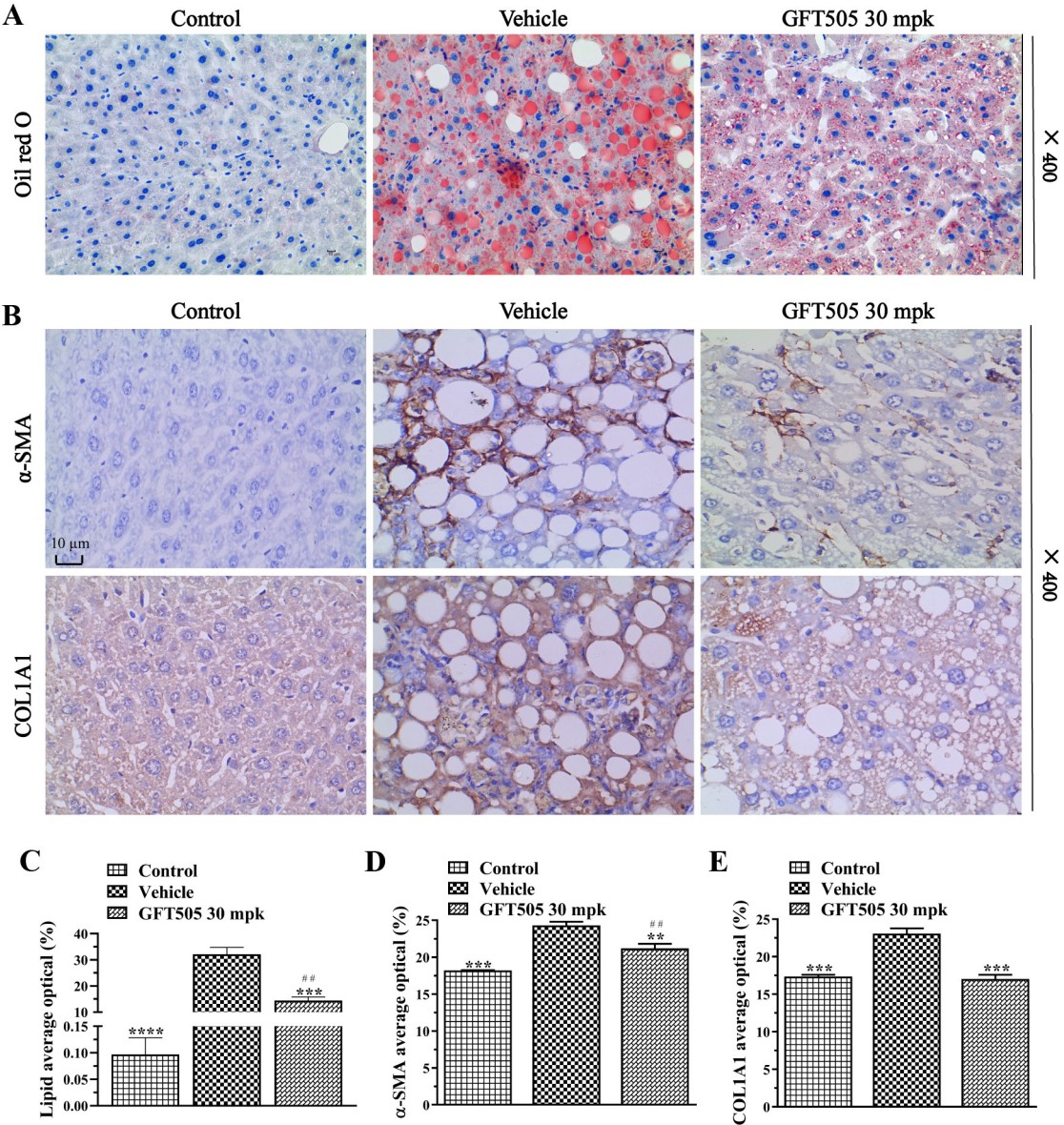

**Fig 3. GFT505 attenuated hepatic lipid accumulation and the expression of α-SMA and collagen I in CDAHFD-fed C57BL/6 mice.** (A) Results of Oil red O staining. (B) Immunohistochemical analysis of α-SMA and collagen I in mice liver sections. (C) Oil red O stained area quantification. (D) α-SMA immunohistochemical area quantification. (E) The collagen I immunohistochemical area quantification. Control, standard diet; Vehicle and GFT505 groups, CDAHFD diet; Statistic analyses were performed using one-way ANOVA; Values were means ± SEM; *$p < 0.05$, **$p < 0.001$, compared with vehicle group; #$p < 0.05$, ##$p < 0.001$, compared with control group; n = 10.

# GFT505 regulated the expression of genes involved in lipid metabolism, inflammation and fibrosis

Total RNA was extracted from the liver tissues of mice in the GFT505 groups, vehicle group. There were 3995 up-regulated genes and 3576 down-regulated genes of 7571 DEGs in GFT505 treatment group compared with vehicle group (Fig 4A). To better understand the transcriptome

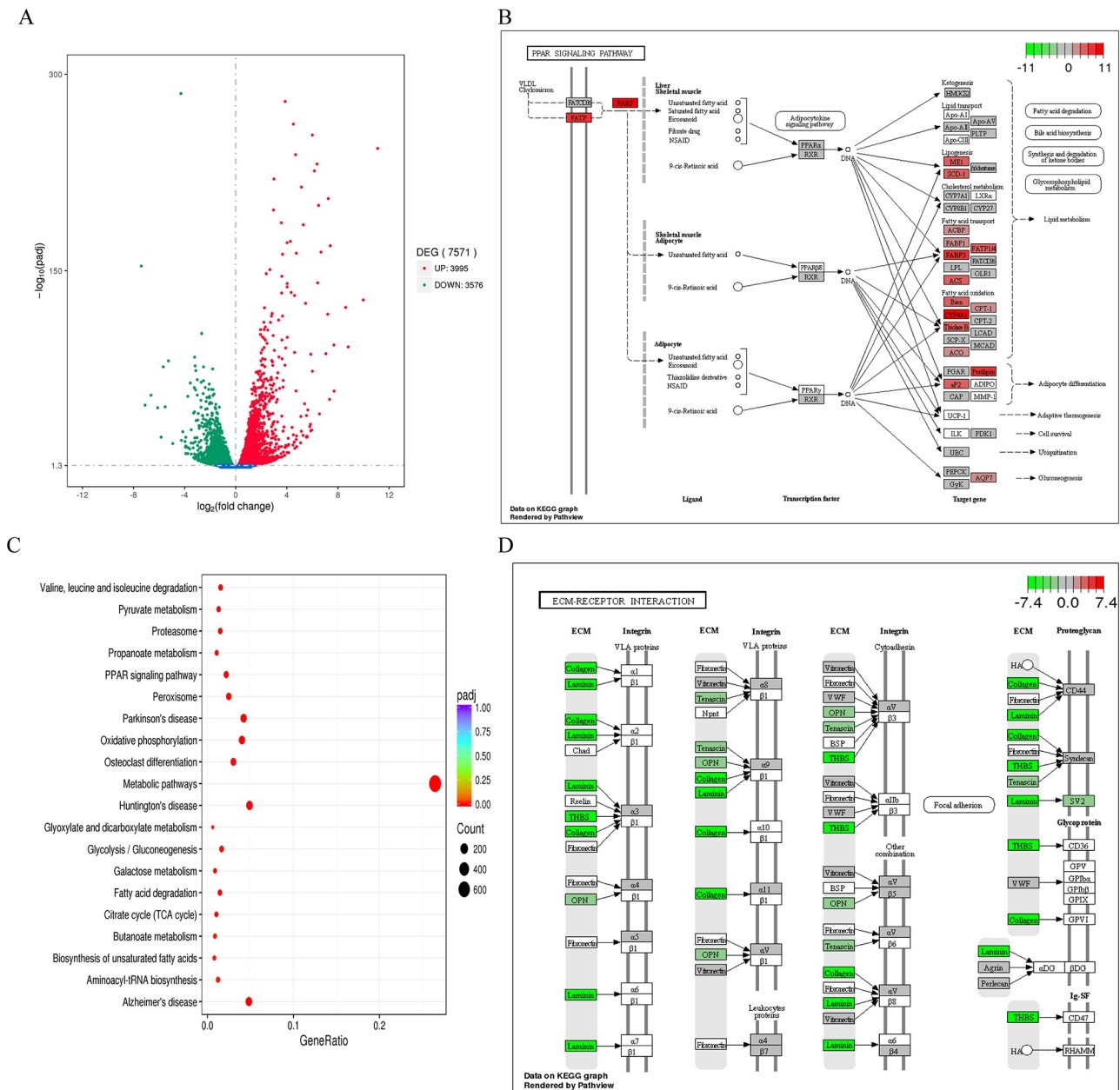

**Fig 4. Effect of GFT505 on the genes expression of PPAR signaling pathway and ECM-receptor interaction pathway.** (A) The differentially expressed genes in GFT505 (30mpk) group, compared with vehicle group. Red dots represent up-regulated DEGs, the green dots represent down-regulated DEGs, and blue dots represent non-DEGs. A total of 7571 unigenes were identified as DEGs ($p < 0.005$ and $\log_2$ [fold change] $> 1$) between treatment and vehicle group. (B) KEGG enrichment analysis of DEGs. Top 20 significantly enriched KEGG pathway. (C) DEGs enriched in "PPAR Signaling pathway". (D), DEGs enriched in "ECM-Receptor interaction". Vehicle and GFT505 group: CDAHFD diet. DEGs, Differentially Expressed Genes; KEGG, Kyoto Encyclopedia of Genes and Genomes; ECM, Extracellular matrix.

regulation of GFT505 treatment in CDAHFD-induced NASH model, we further performed Gene ontology (GO) analysis and Kyoto Encyclopedia of Genes and Genomes (KEGG) pathway analysis to classify DEGs into relevant functions. The results of ten most significant Biological Process by GO analysis were listed in S1 Table. The lipid metabolism processes, such as fatty acid metabolic process and lipid biosynthetic process, were enriched. The results of the Reactom enrichment analysis is similar to GO analysis (S2 Table). The top 10 enriched KEGG pathways were presented in Fig 4B and listed in S3 Table. Among which the enriched PPAR signaling pathway and Fatty acid degradation pathway were showed in Fig 4C and S1 Fig. Besides, the ECM-receptor interaction pathway (Fig 4D) and Cytokine-cytokine receptor interaction pathway (S2 Fig) were enriched as well. As shown in S1 Fig, *Ehhadh* and *Acaa2* were up-regulated in fatty acid degradation pathway. And the Cytokine-cytokine receptor interaction pathway genes involved in inflammation were down-regulated, such as *Cxcl1*, *Cxcl2*, *Cxcl5*, *Cxcl4*, *Ccl21*, *Ccl22*, *Il6r*, *Il7r*, *Tnf* and *Ccr3*. As for ECM-receptor interaction pathway, *Collagen* and *Laminin* were significantly down-regulated. Furthermore, heat map was made to present DEGs related to lipid metabolism, inflammation and fibrosis (Fig 5A), and RT-qPCR was performed to confirmed these related mRNA expressions (Fig 5B and 5D). The changes of the related genes expression of lipid metabolism, inflammation reaction and fibrosis in RNA-SEQ or qRT-PCR were showed in S4 Table. The primers sequences for RT-qPCR were showed in S5 Table. In summary, GFT505 increased the expression of genes involved in lipid metabolism and decreased inflammation and fibrosis related gene expression in CDAHFD-induced NASH model.

## GFT505 inhibited lipid accumulation in LO2 cell

To further validate the effect of GFT505 treatment in lipid accumulation, the human normal liver cell LO2 was treated with palmitic and oleate acid for 24 h. After treating with palmitic and oleate acid, LO2 cell showed massive lipid droplets accumulation in vehicle group. The lipid accumulation was alleviated by GFT505 in a dose-dependent manner (Fig 6A). The quantification of Oil red O staining was showed in Fig 6B. The analysis of TG concentration was showed in Fig 6C and the result was consistent with Oil red O staining. In conclusion, GFT505 treatment reduced lipid accumulation through LO2 cell Oil red O staining and TG concentration analysis in vitro.

## Discussion

NASH is a progressive disease usually accompanied by metabolic syndrome, insulin resistance and glucose tolerance. The evaluation of therapeutic effects about drugs in NASH should include insulin resistance, steatosis, inflammation, oxidative stress, mitochondrial dysfunction and fibrosis. PPARs, a family of ligand-activated nuclear receptors, are highly expressed in the liver and adipose tissue regulating the lipid and glucose metabolism. Several specific subtype-selective PPARs agonists have been applied to reduce experimental fibrotic steatohepatitis [14–16]. The efficacy of PPAR α/δ dual agonist GFT505 had been evaluated in several animal models previously, including western diet-fed human APOE2 transgenic mice, MCD diet-fed db/db mice and CCL4-induced fibrosis rats [9, 17]. In this study, we firstly evaluated the pharmacodynamics of PPAR α/δ agonist GFT505 and profile the regulatory network of GFT505 in the CDAHFD induced NASH model.

Many diets induced NASH animal model were reported. They replicate different characteristic of human NASH [18]. High-fat diet (HFD) model showed similar features with human NASH such as obesity, IR, steatosis, inflammation, but the hepatic pathologic degree is mild and takes longer than other diet models [19, 20]. Methionine-choline-deficient diet (MCD) induces steatosis, hepatic inflammation and fibrosis in a short feeding time [18, 21]. However,

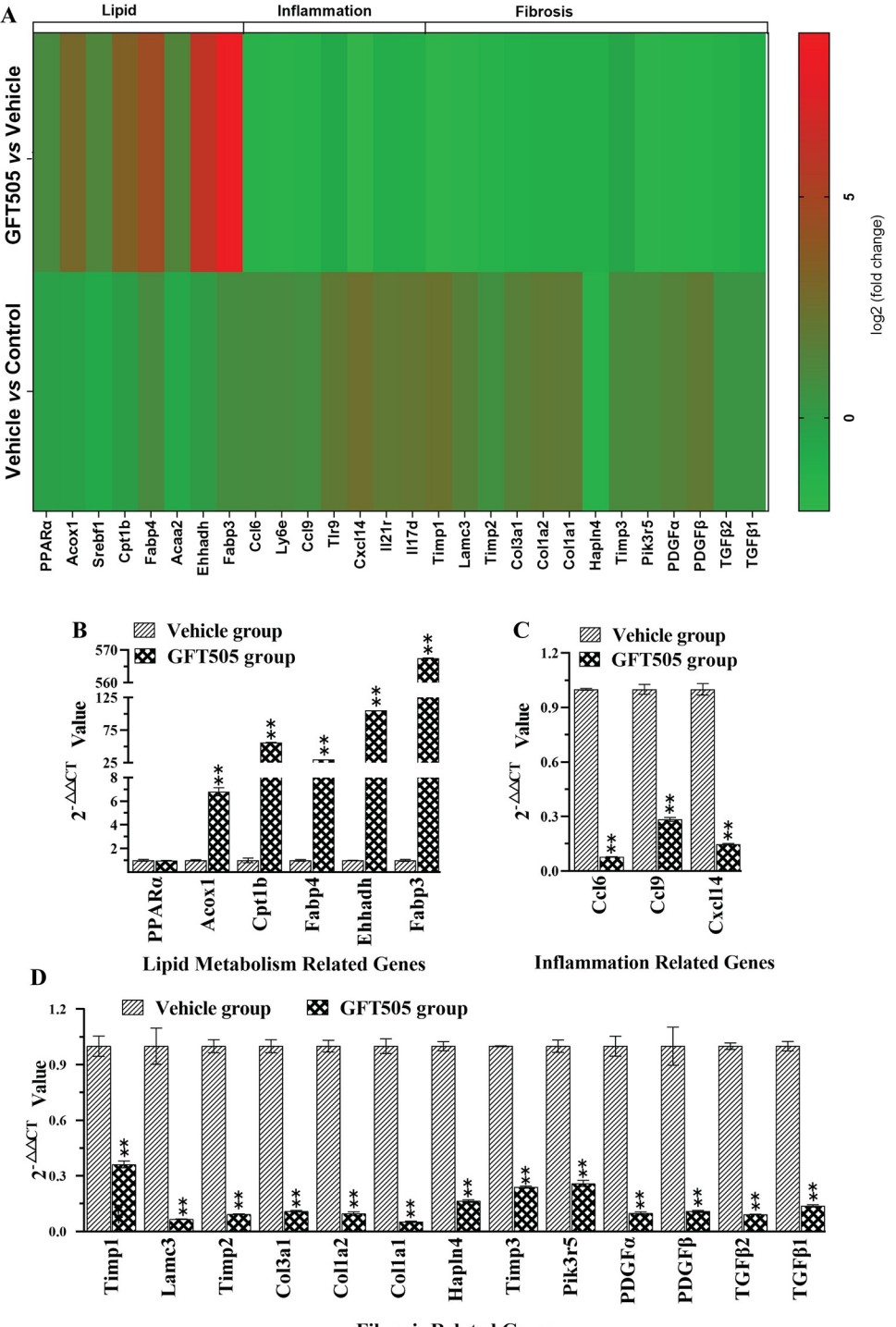

**Fig 5. The different genes expression about lipid metabolism, inflammation reaction, and fibrosis after GFT505 30 mpk treatment.** (A) The hot map of lipid related-, inflammation related-, and fibrosis related-DEGs. (B, C, D) The mRNA expression of lipid metabolism related- and inflammation related-DEGs were confirmed by RT-qPCR. Vehicle and GFT505 group: CDAHFD diet. DEGs, Differentially Expressed Genes. Statistic analyses were performed using one-way ANOVA; Values were means ± SEM; $^{**}p < 0.0001$, comparing with vehicle group; $^{##}p < 0.0001$, comparing with control group; n = 3.

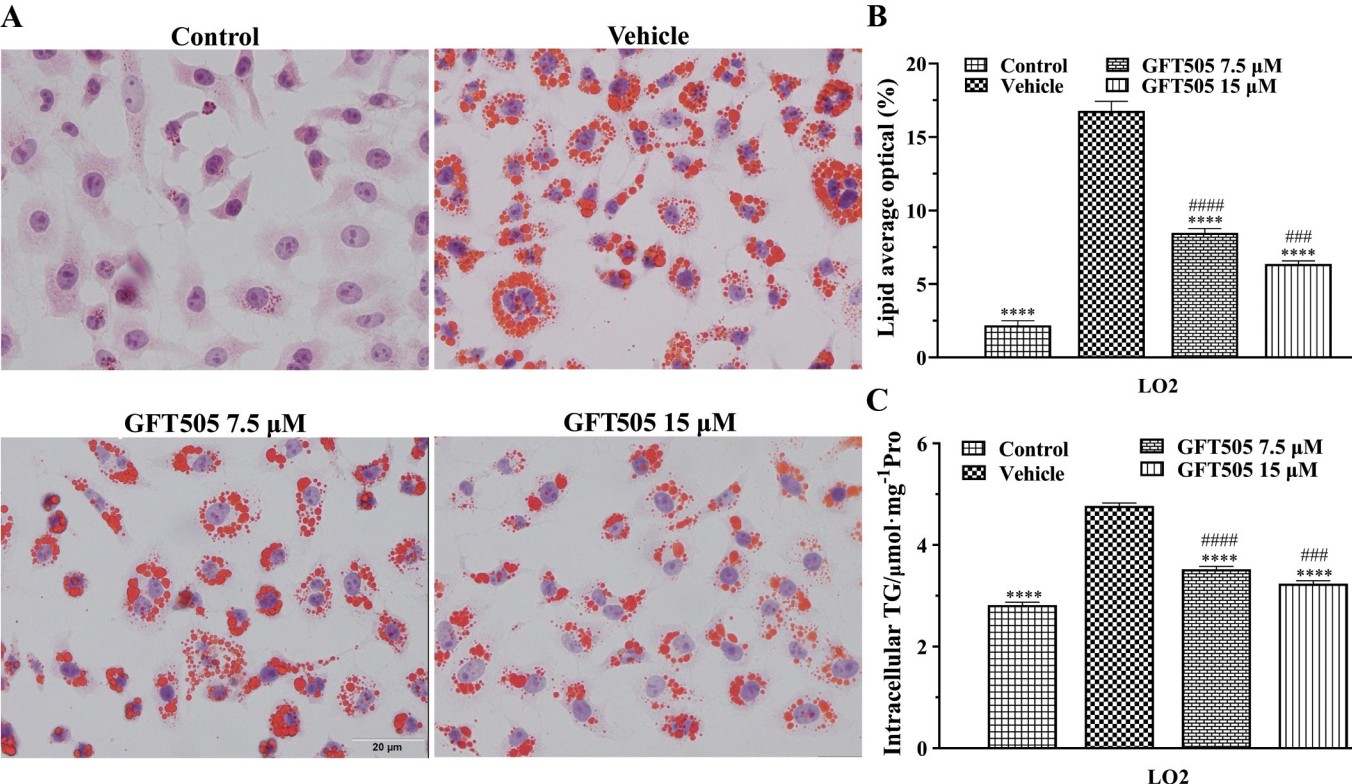

**Fig 6. GFT505 reduced lipid accumulation in LO2 cell.** (A) GFT505 reduced lipid accumulation in LO2 cell by Oil red O staining. (B) The quantification of Oil red O staining. (C) TG content was quantified by microplate reader in LO2 cell. Statistical analyses were performed using one-way ANOVA; Values were presented as means ± SEM; ****$p < 0.0001$, comparing with vehicle group; ####$p < 0.0001$, comparing with control group; n = 3.

this model is usually not accompanied by obesity and peripheral IR. Moreover, the decreased body weight limits the use of MCD model [22, 23]. The choline-deficient, L-amino acid-defined (CDAA) model is similar to MCD model except for more serious histological changes [18]. The choline-deficient, L-amino acid-defined, high-fat diet (CDAHFD) model was established by the combination of CDAA and HFD models. CDAHFD mouse model can not only simulate human NASH disease to develop steatosis, inflammation and fibrosis in a short period, but also overcome the body weight loss [24]. Therefore, the novel CDAHFD mouse model was selected in our study to investigate the PD and mechanism of PPAR α/δ dual agonist GFT505.

In CDAHFD animal model, the levels of serum ALT and AST increased with hepatomegaly. Besides, serum levels of TG in vehicle group were lower than that in control group GFT505. The histology analysis on steatosis (H&E staining), fibrosis (SR staining) and α-SMA and collagen I staining (immunohistochemical analysis) of liver tissue demonstrated that GFT505 can alleviate steatosis, inflammation and fibrosis. In brief, PPAR α/δ agonist GFT505 improved liver function, prevented intrahepatic lipid accumulation and reduced liver inflammation and fibrosis in CDAHFD animal model.

Furthermore, high-throughput comparative RNA-Seq analysis was used to evaluate the gene expression profiles after GFT505 treatment. High-throughput sequence and DEGs analysis showed that there were 3995 up-regulated genes and 3576 down-regulated genes in GFT505 treatment group. Lipid metabolism related genes *Cpt1b*, *Acox1*, *Ehhadh*, *Fabp3*, *Fabp4*, *Srebf1* *Acaa2* and *Pparα* were up-regulated compared with the vehicle group. The down-regulation of

*Cpt1b* and *Acox1* genes could lead to lipid accumulation in liver, thus slowing down fatty acid synthesis and β-oxidation [25, 26]. Down-regulation of *Ehhadh* gene expression decreased very low density lipoprotein (VLDL) transport rate [27]. *Pparα* is involved in free fatty acid transport, oxidation, liver lipids and plasma lipoprotein metabolism [28]. *Fabp4* is highly expressed in adipocytes and also exists in macrophages and dendritic cells, which is related to diseases such as obesity and insulin resistance. A recent study confirmed that in *Fabp4* knockout mice, the IKK (IκB kinase) and NF-κB (nuclear factor-κB) activities were inhibited and the accumulation of cholesterol, fatty acids and lipid droplets was reduced, showing the reduced inflammation [29]. GFT505 could upregulate the expression of *Fabp4* gene, thus regulating fat metabolism and inflammation. The effect of changes in *Fabp4* gene expression on CDAHFD model mice deserves further study. Changes in lipid metabolism related genes led to significant enrichment of fatty acid metabolic related processes and upregulation of fatty acid degradation pathway. Furthermore, the effect of GFT505 on ameliorating fat accumulation was verified on LO2 cell model *in vitro*. A short-term phase II clinical study showed that GFT505 reduced plasma triglyceride levels, increased high-density lipoprotein cholesterol levels, and decreased low-density lipoprotein cholesterol serum levels in prediabetic patients [10]. Besides, inflammation-related genes such as *Ly6e*, *Tlr9* and *Il17d* were also down-regulated. Among them, *Ly6e* (lymphocyte antigen 6 complex, locus E) plays an important role in regulating immunity, T cell physiological characteristics and tumor formation and the increased expression of *Ly6e* lead to the increased risk of human infection with the virus [30]. *Il17d* plays a key role in the recruitment of human immune cells [31]. Studies have reported that *Tlr9* caused acute injury by activating IL-17A [32]. In addition, the expression of genes such as *Ccl6* (*C-C Motif Chemokine Receptor 6*), *Ccl9* (*C-C Motif Chemokine Receptor 9*), *Cxcl14* (*C-X-C Motif Chemokine Ligand 14*) and *Il21r* (*interleukin 21 receptor*) were also significantly down-regulated after treatment with GFT505. This may contribute to the decreased inflammation. Due to the downregulation of inflammation-related genes expression, cytokine-cytokine receptor interaction pathway was also down-regulated. Furthermore, fibrosis-related genes *Timp1*, *Lamc3* (*Laminin Subunit Gamma3*), *Timp2*, *Col3a1*, *Col1a2*, *Col1a1*, *Hapln4* (*Hyaluronan And Proteoglycan Link Protein 4*), *Timp3*, *Pik3r5*, *Pdgfα*, *Pdgfβ*, *Tgfβ1* and *Tgfβ2* were significantly down-regulated, which caused the downregulation of ECM-receptor interaction pathway after GFT505 treatment.

In summary, the PPAR α/δ dual agonist GFT505 showed efficacy on improving serum biochemistry and alleviating steatosis, inflammation and fibrosis of liver at both protein and gene expression levels in CDAHFD animal model. Lipid metabolism-related genes *Cpt1b*, *Acox1*, *Ehhadh*, *Fabp3*, *Srebf1* and *Acaa2*, inflammation-related genes *Ly6e*, *Tlr9*, *Il17d*, *Ccl6*, *Ccl9*, *Cxcl14* and *Il21r*, and fibrosis-related genes *Timp1*, *Lamc3*, *Timp2*, *Col3a1*, *Col1a2*, *Col1a1*, *Hapln4*, *Timp3*, *Pik3r5*, *Pdgfα*, *Pdgfβ*, *Tgfβ1* and *Tgfβ2* regulated by GFT505 are identified *in vivo*. This study further confirmed the effect of PPAR α/δ dual agonist in NASH, and also provided clues for further elucidating the mechanism of PPAR α/δ agonists in the treatment of NASH. However, not considering the effects of GFT505 on normal mice was a major limitation of our study design. We will further explore the influence of GFT505 on control group.

## Supporting information

**S1 Fig. GFT505 Treatment up-regulates "fatty acid degradation" pathway related genes.**
(TIF)

**S2 Fig. GFT505 Treatment down-regulates "cytokine-cytokine receptor interaction" pathway related genes.**
(TIF)

**S1 Table. GFT505 treatment most significantly affect 10 biological process by Gene Ontology enrichment analysis.**
(DOC)

**S2 Table. GFT505 treatment most significantly affect 10 process by Reactom enrichment analysis.**
(DOC)

**S3 Table. GFT505 treatment most significantly affect 10 pathways by Kyoto encyclopedia of genes and genomes enrichment analysis.**
(DOC)

**S4 Table. The change of lipid metabolism related genes, inflammation reaction related genes and fibrosis related genes expression.**
(DOC)

**S5 Table. Primers sequences used for RT-qPCR.**
(DOC)

## Acknowledgments

The authors thank the research platform provided by Public Health and Preventive Medicine Provincial Experiment Teaching Center at Sichuan University and Food Safety Monitoring and Risk Assessment Key Laboratory of Sichuan.

## Author Contributions

**Data curation:** Linfu Liu, Chuang Liu, Manyu Zhao, Qianru Zhang, Ying Lu.

**Formal analysis:** Linfu Liu, Chuang Liu, Manyu Zhao, Qianru Zhang, Ying Lu.

**Funding acquisition:** Jinliang Yang, Xiaoxin Chen, Yuqin Yao.

**Investigation:** Linfu Liu, Chuang Liu, Qianru Zhang.

**Methodology:** Chuang Liu, Qianru Zhang, Yuqin Yao.

**Project administration:** Ping Liu, Jinliang Yang, Yuqin Yao.

**Resources:** Ping Liu, Hua Yang, Jinliang Yang, Xiaoxin Chen, Yuqin Yao.

**Software:** Chuang Liu, Qianru Zhang.

**Supervision:** Hua Yang, Jinliang Yang, Xiaoxin Chen, Yuqin Yao.

**Visualization:** Linfu Liu, Chuang Liu, Manyu Zhao, Qianru Zhang, Ying Lu, Yuqin Yao.

**Writing – original draft:** Linfu Liu, Chuang Liu, Qianru Zhang.

**Writing – review & editing:** Linfu Liu, Chuang Liu, Manyu Zhao, Qianru Zhang, Ying Lu, Yuqin Yao.

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
