## [Decision Letter · Decision Letter 0]

8 Jun 2020

PONE-D-20-10913

The pharmacodynamic and differential gene expression analysis of PPAR α/δ agonist GFT505 in CDAHFD-induced NASH model

PLOS ONE

Dear Dr. Yao,

Thank you for submitting your manuscript to PLOS ONE. After careful consideration, we feel that it has merit but does not fully meet PLOS ONE’s publication criteria as it currently stands. Therefore, we invite you to submit a revised version of the manuscript that addresses the points raised during the review process.

Both of the reviewers believed the study was interesting and a valuable addition to the literature.  However there were a number of issues that need to be address as listed by the reviewers.  Additionally, the The RNAseq datasets should be put in a database to be readily accessed upon publication and their were a number of grammatical issues that need to be addressed.

We look forward to receiving your revised manuscript.

Kind regards,

Jonathan M Peterson, Ph.D.

Academic Editor

PLOS ONE

Journal Requirements:

2. We note that you are reporting an analysis of a microarray, next-generation sequencing, or deep sequencing data set. PLOS requires that authors comply with field-specific standards for preparation, recording, and deposition of data in repositories appropriate to their field. Please upload these data to a stable, public repository (such as ArrayExpress, Gene Expression Omnibus (GEO), DNA Data Bank of Japan (DDBJ), NCBI GenBank, NCBI Sequence Read Archive, or EMBL Nucleotide Sequence Database (ENA)). In your revised cover letter, please provide the relevant accession numbers that may be used to access these data. For a full list of recommended repositories, see http://journals.plos.org/plosone/s/data-availability#loc-omics or http://journals.plos.org/plosone/s/data-availability#loc-sequencing.

3. Thank you for including your competing interests statement; "The authors have declared that no competing interests exist."

We note that one or more of the authors are employed by a commercial company: Guangdong Zhongsheng Pharmaceutical Co., Ltd

4. To comply with PLOS ONE submission requirements, in your Methods section, please provide additional information regarding the experiments involving animals and ensure you have included details on (1) methods of sacrifice, (2) methods of anesthesia and/or analgesia, and (3) efforts to alleviate suffering.

Additional Editor Comments (if provided):

Both of the reviewers believed the study was interesting and a valuable addition to the literature. However there were a number of issues that need to be address as listed by the reviewers. Additionally, the The RNAseq datasets should be put in a database to be readily accessed upon publication and their were a number of grammatical issues that need to be addressed.

Reviewers' comments:

Reviewer's Responses to Questions

**Comments to the Author**

1. Is the manuscript technically sound, and do the data support the conclusions?

Reviewer #1: Yes

Reviewer #2: Partly

2. Has the statistical analysis been performed appropriately and rigorously? 

Reviewer #1: Yes

Reviewer #2: Yes

3. Have the authors made all data underlying the findings in their manuscript fully available?

Reviewer #1: No

Reviewer #2: Yes

4. Is the manuscript presented in an intelligible fashion and written in standard English?

Reviewer #1: Yes

Reviewer #2: Yes

5. Review Comments to the Author

Reviewer #1: The authors assessed the effects of the PPARa/d dual agonist GFT505 (elafibranor) on liver histology and gene expression in a mouse model of NASH and fibrosis induced by a choline-deficient diet.

Remarks:

-GFT505 is now termed elafibranor. The authors may want to use its generic name.

-Since choline deficiency impacts on hepatic VLDL output, the impact of GFT505 on plasma lipid and lipoprotein parameters is difficult to evaluate. Moreover, it is unclear how the authors assessed plasma HDL and LDL. This reviewer assumes it is cholesterol in these lipoproteins that was measured? If this was done using kits for lipoprotein cholesterol and triglyceride measurement in humans, the results may incorrectly be termed LDL-C and HDL-C. The authors should either eliminate these data from the paper or perform more detailed cholesterol and triglyceride distribution profile analysis on plasma of the animals in order to have a clearer view on the plasma lipid changes.

-The effects of PPARa agonism on liver mass are due to peroxisome proliferation (erroneously mentioned as liver swelling). This only occurs in rodents and therefore should not be classified as side effect.

Reviewer #2: The manuscript entitled "The pharmacodynamic and differential gene expression analysis of PPAR α/δ agonist GFT505 in CDAHFD-induced NASH model" by Liu et al. demonstrates that GFT505 reduces lipid accumulation, inflammation and fibrosis in a NASH mouse model. The histological data is clear. However, the manuscript presents several issues:

Major comments:

1. This study lacks a mechanism by which GFT505 reduces fat accumulation in hepatocytes, inflammation and fibrosis.

2. Relevance to human disease is missing.

3. The manuscript has many English typos.

Specific comments:

1. Abstract: please revise the sentence starting with “The significant regulation genes…”, it is quite confusing.

2. The authors should revise the English. Example: Introduction, line 9, please replace “PPAR α, which mainly expresses…” by “PPAR α, which is mainly expressed”.

3. Why only male mice are used in this study? The authors should also use female mice.

4. When authors state that mice were divided in 5 groups, does the n=10 mean 10 mice/group or 10 mice in total? Please specify.

5. Figure 1A: why vehicle decreases the body weight compared to control?

6. The authors should specify in the figure panels which groups received normal diet and which received CDAHFD.

7. Figure legends are not very explanatory. They should describe briefly the experiment for each panel.

8. Figure 3: it would be nice to confirm these results by performing WB, which gives a better idea on the levels of Col1a1 and aSMA.

9. Figure 4 is hard to read, the letters are too small and the resolution of the figure is not ideal.

10. Does the target genes show the same trend in human NASH livers compared to healthy individuals as in the NASH model?

11. What types of inflammatory cells are reduced after GFT505 treatment?

12. In this study, is GFT505 used as a preventive or a curative treatment?

13. What are the effects of GFT505 in control diet-fed mice?

6. PLOS authors have the option to publish the peer review history of their article (what does this mean?). If published, this will include your full peer review and any attached files.

Reviewer #1: No

Reviewer #2: No

---

## [Author Response · Author response to Decision Letter 0]

27 Aug 2020

Dear Editor and Reviewers, 

Thanks for your professional and valuable comments on the manuscript (PONE-D-20-10913) entitled "The pharmacodynamic and differential gene expression analysis of PPAR α/δ agonist GFT505 in CDAHFD-induced NASH model". We have carefully addressed the editors’ and reviewers’ comments and revised the manuscript. The point-to-point corrections we did are following.

Criticisms: The point-to-point corrections we did are following.

Editor’s advice:

1. Both of the reviewers believed the study was interesting and a valuable addition to the literature. However, there were a number of issues that need to be address as listed by the reviewers. Additionally, the RNAseq datasets should be put in a database to be readily accessed upon publication and there were a number of grammatical issues that need to be addressed.

Reply: thanks for your advice. We have uploaded our RNAseq data to NCBI Sequence Read Archive and have revised the grammar errors throughout the manuscript.

2. We note that you are reporting an analysis of a microarray, next-generation sequencing, or deep sequencing data set. PLOS requires that authors comply with field-specific standards for preparation, recording, and deposition of data in repositories appropriate to their field. Please upload these data to a stable, public repository (such as ArrayExpress, Gene Expression Omnibus (GEO), DNA Data Bank of Japan (DDBJ), NCBI GenBank, NCBI Sequence Read Archive, or EMBL Nucleotide Sequence Database (ENA)). In your revised cover letter, please provide the relevant accession numbers that may be used to access these data. For a full list of recommended repositories, see http://journals.plos.org/plosone/s/data-availability#loc-omics or http://journals.plos.org/plosone/s/data-availability#loc-sequencing.

Reply: thanks for your suggestions. We have uploaded the data to NCBI Sequence Read Archive and the relevant accession number is PRJNA657593.

3. Thank you for including your competing interests statement; "The authors have declared that no competing interests exist."

We note that one or more of the authors are employed by a commercial company: Guangdong Zhongsheng Pharmaceutical Co., Ltd

Please provide an amended Funding Statement declaring this commercial affiliation, as well as a statement regarding the Role of Funders in your study. If the funding organization did not play a role in the study design, data collection and analysis, decision to publish, or preparation of the manuscript and only provided financial support in the form of authors' salaries and/or research materials, please review your statements relating to the author contributions, and ensure you have specifically and accurately indicated the role(s) that these authors had in your study. You can update author roles in the Author Contributions section of the online submission form. Please also include the following statement within your amended Funding Statement“The funder provided support in the form of salaries for authors [insert relevant initials], but did not have any additional role in the study design, data collection and analysis, decision to publish, or preparation of the manuscript. The specific roles of these authors are articulated in the ‘author contributions’ section.” Please also provide an updated Competing Interests Statement declaring this commercial affiliation along with any other relevant declarations relating to employment, consultancy, patents, products in development, or marketed products, etc. Within your Competing Interests Statement, please confirm that this commercial affiliation does not alter your adherence to all PLOS ONE policies on sharing data and materials by including the following statement: "This does not alter our adherence to PLOS ONE policies on sharing data and materials.” (as detailed online in our guide for authors http://journals.plos.org/plosone/s/competing-interests). If this adherence statement is not accurate and there are restrictions on sharing of data and/or materials, please state these. Please note that we cannot proceed with consideration of your article until this information has been declared.

Reply: thanks for your suggestion. We have updated the competing interest statement, author contributions and funding statement. Moreover, we have added competing interest statement and funding statement in the manuscript.

Although there were two authors employed by Guangdong Zhongsheng Pharmaceutical Co., Ltd involved in our research, the company didn’t fund the study. The specific roles of these authors are articulated in the ‘author contributions’ section.” We fell so sorry for not explaining the conflict of interests about this. 

Funding Statement

This work was supported by the National Major Scientific and Technological Special Project (Nos. 2018ZX09201002, 2018ZX09711001-011 and 2019ZX09201001) and the National Science Foundation of China (No. 81773375). The authors thank the research platform provided by Public Health and Preventive Medicine Provincial Experiment Teaching Center at Sichuan University.

Competing interest statement

All the authors declare that they have no conflict of interests. And all the institutions and companies declare that they have no conflict of interests.

In this study, Sichuan University, including Collaborative Innovation Center for Biotherapy and Cancer Center, West China School of Public Health and West China Fourth Hospital, provided platforms for performing the experiments, but did not have any additional role in the study design, data collection and analysis, decision to publish, or preparation of the manuscript. And Guangdong Zhongsheng Pharmaceutical Co., Ltd does not fund the study and alter our adherence to PLOS ONE policies on sharing data and materials. The specific roles of the authors were articulated in the ‘author contributions’ section.

Author contributions

Qianru Zhang, Manyu Zhao, Linfu Liu and Chuang Liu performed most of the experiments. Ying Lu and Hua Yang collected and interpreted data. Linfu Liu and Chuang Liu wrote this paper. Qianru Zhang, Yuqin Yao and Jinliang Yang revised the paper. Yuqin Yao, Jinliang Yang, and Xiaoxin Chen were responsible for fund collection and revised this paper. Qianru Zhang and Xiaoxin Chen are employees of Guangdong Zhongsheng Pharmaceutical Co., Ltd.. Qianru Zhang is an on-the-job ph.d.student of Sichuan University 

Reply: thanks for your advice. We have updated Funding Statement and Competing Interests Statement in our cover letter.

5. To comply with PLOS ONE submission requirements, in your Methods section, please provide additional information regarding the experiments involving animals and ensure you have included details on (1) methods of sacrifice, (2) methods of anesthesia and/or analgesia, and (3) efforts to alleviate suffering.

Reply: thanks for your suggestions. Animals were anesthetized with 3% pentobarbital sodium by intraperitoneal injection. And then the terminal blood samples were obtained from the eyelids for biochemical analysis (mice were sacrificed in this way). We have added these details in the section of “CDAHFD-fed mouse NASH model”.

Reviewers' comments:

Reviewer #1: The authors assessed the effects of the PPARa/d dual agonist GFT505 (elafibranor) on liver histology and gene expression in a mouse model of NASH and fibrosis induced by a choline-deficient diet.

Remarks:

-GFT505 is now termed elafibranor. The authors may want to use its generic name.

Reply: thanks for your review. We have descibed the generic name of GFT505 in abstract.

-Since choline deficiency impacts on hepatic VLDL output, the impact of GFT505 on plasma lipid and lipoprotein parameters is difficult to evaluate. Moreover, it is unclear how the authors assessed plasma HDL and LDL. This reviewer assumes it is cholesterol in these lipoproteins that was measured? If this was done using kits for lipoprotein cholesterol and triglyceride measurement in humans, the results may incorrectly be termed LDL-C and HDL-C. The authors should either eliminate these data from the paper or perform more detailed cholesterol and triglyceride distribution profile analysis on plasma of the animals in order to have a clearer view on the plasma lipid changes.

Reply: thanks for your kind comments. The serum HDL levels was detected by HDL-Cholesterol Gen.4 (Roche, Basel, Switzerland) and serum LDL levels was detected by LDL-Cholesterol Gen.3 (Roche, Basel, Switzerland). Considering your suggestion, we have eliminated these data.

-The effects of PPARa agonism on liver mass are due to peroxisome proliferation (erroneously mentioned as liver swelling). This only occurs in rodents and therefore should not be classified as side effect.

Reply: sincerely thanks for your useful suggestion. We have deleted this desciption.

Reviewer #2: The manuscript entitled "The pharmacodynamic and differential gene expression analysis of PPAR α/δ agonist GFT505 in CDAHFD-induced NASH model" by Liu et al. demonstrates that GFT505 reduces lipid accumulation, inflammation and fibrosis in a NASH mouse model. The histological data is clear. However, the manuscript presents several issues:

Major comments:

1. This study lacks a mechanism by which GFT505 reduces fat accumulation in hepatocytes, inflammation and fibrosis.

Reply: thanks for your kind comments. The objective of our study was to profile the regulatory network of PPAR α/δ agonist in NASH and test the pharmacodynamics of elafibranor. And we found that GFT505 could regulate the genes involved in glucolipid metabolism (Pparα, Acox1, Cpt1b, Fabp4, Ehhadh, Fabp3), inflammation (Ccl6, Ccl9, Cxcl14) and fibrosis (Timp1, Lamc3, Timp2, Col3a1, Col1a2, Col1a1, Hapln4, Timp3, Pik3r5, Pdgfα, Pdgfβ, Tgfβ1, Tgfβ2), which were confirmed by qRT-PCR. The down-regulated genes were significantly enriched in cytokine-cytokine receptor interaction pathway and ECM-receptor interaction pathway, while the up-regulated genes were enriched in PPAR signaling pathway and fatty acid degradation pathway. These results suggest that GFT505 may play a role in inhibiting fat accumulation, inflammation and fibrosis in NASH by regulating these genes and signaling pathways. And we will futher investigate the exact mechanism of GFT505 on NASH regualting the specific gene or signaling pathway. Thanks for your useful comments again.

2. Relevance to human disease is missing.

Reply: thanks for your review. NASH is a progressive disease from NAFLD, which characterized by steatosis, hepatocellular ballooning, lobular inflammation and fibrosis. Because of the slow progress of NASH, although the incidence rate is increasing, the overall incidence rate is still low, and clinical samples are difficult to obtain. And our study is only a preclinical study of NASH and the aims of our study is to investigate the pharmacodynamics and transcriptome regulation of GFT505 in vivo and in vitro. 

3. The manuscript has many English typos.

Reply: thanks for your advice. We have correted our grammatical errors throughout the manuscript.

Specific comments:

1. Abstract: please revise the sentence starting with “The significant regulation genes…”, it is quite confusing.

Reply: thanks for your advice. We have revised this sentence.

2. The authors should revise the English. Example: Introduction, line 9, please replace “PPAR α, which mainly expresses…” by “PPAR α, which is mainly expressed”.

Reply: thanks for your carefully review. We have revised the grammatical errors in our manuscript, e.g., “PPAR α, which mainly expresses…” have been corrected into “PPAR α, which is mainly expressed”.

3. Why only male mice are used in this study? The authors should also use female mice.

Reply: thanks for your advice. We chose male mice from the effectiveness of the NASH model. Studies have shown that estrogens can protect female mice from liver steatosis and fibrosis in female mice(1). Moreover, female specific white adipose tissue browning helps protect female mice from fatty liver disease induced by MCD diet(2). 

Reference: 

1. Ballestri S, Nascimbeni F, Baldelli E, Marrazzo A, Romagnoli D, Lonardo A. NAFLD as a Sexual Dimorphic Disease: Role of Gender and Reproductive Status in the Development and Progression of Nonalcoholic Fatty Liver Disease and Inherent Cardiovascular Risk. Advances in therapy. 2017;34(6):1291-326.

2. Lee YH, Kim SH, Kim SN, Kwon HJ, Kim JD, Oh JY, et al. Sex-specific metabolic interactions between liver and adipose tissue in MCD diet-induced non-alcoholic fatty liver disease. Oncotarget. 2016;7(30):46959-71.

4. When authors state that mice were divided in 5 groups, does the n=10 mean 10 mice/group or 10 mice in total? Please specify.

Reply: thanks for your suggestion. We are so sorry for not describing animal amounts clearly. “n=10” means 10 mice in each group. We have specified this in the section of “CDAHFD-fed mouse NASH model”.

5. Figure 1A: why vehicle decreases the body weight compared to control?

Reply: in our study, Model group was given CDAHFD, GFT505 groups were given GFT505 and CDAHFD, control group was given normal diet. GFT505 was dissolved by vehicle. The 1% carboxymethyl cellulose sodium and 0.1% tween 80 are the components of vehicle, which have been widely used as solvent of drugs(1,2) and studies have shown that they have no obvious effect on mice weight(2). Compared with the control group, the body weight of vehicle and GFT505 groups decreased, and there was no difference between GFT505 groups and model group, which mainly indicated that bleomycin caused the weight loss of mice.

Reference:

1. Camilleri C, Beiter RM, Puentes L, Aracena-Sherck P, Sammut S. Biological, Behavioral and Physiological Consequences of Drug-Induced Pregnancy Termination at First-Trimester Human Equivalent in an Animal Model. Frontiers in neuroscience. 2019;13:544.

2. Chassaing B, Koren O, Goodrich JK, Poole AC, Srinivasan S, Ley RE, et al. Dietary emulsifiers impact the mouse gut microbiota promoting colitis and metabolic syndrome. Nature. 2015;519(7541):92-6.

6. The authors should specify in the figure panels which groups received normal diet and which received CDAHFD.

Reply: thanks for your suggestions. We have specified which groups received normal diet and which received CDAHFD both in methods section and figure legends.

7. Figure legends are not very explanatory. They should describe briefly the experiment for each panel.

Reply: thanks for your kind comments. We have revised the figure legends and described the experiment for each panel.

8. Figure 3: it would be nice to confirm these results by performing WB, which gives a better idea on the levels of Col1a1 and aSMA.

Reply: thanks for your professional comments. The WB results of Col1a1 and a-SMA were shown in R-figure1, which is writen in the rebuttal letter. The protein level of Collagen 1 in GFT505 group was lower than that in the Vehicle group, but there was no statistical significance. And the level of α-SMA was decreased by GFT505 treatment compared with Vehicle group. The WB results of Collagen 1 was not consistent with the results of immunohistochemistry. The reason may be that there is steatosis in NASH animal model, and fat has a great influence on the extraction of tissue protein, which will lead to the loss of two proteins. In addition, Collagen 1 is a kind of secreted protein, which is not suitable for Western blot detection and is easy to degrade (our tissue samples were extracted in 2017). In addition, immunohistochemistry and Sirius Red staining are the standard methods for fibrosis evaluation.

R-figure 1. The protein expression levels of Collagen 1 and α-SMA. A. The results of the protein expression levels of Collagen 1 and α-SMA detected by western blotting. B. Quantification of A (Collagen 1). C. Quantification of A (α-SMA).

9. Figure 4 is hard to read, the letters are too small and the resolution of the figure is not ideal.

Reply: thanks for you review. We have uploaded the high resolution of figure4.

10. Does the target genes show the same trend in human NASH livers compared to healthy individuals as in the NASH model?

Reply: thanks for your review. Our study is a preclinical study of NASH. The samples used for RNA-sequecing were obtained from mouse tissue samples. We will futher investigate whether these target genes showing the same trend in human NASH.

11. What types of inflammatory cells are reduced after GFT505 treatment?

Reply: thanks for your professional comments. This is a shortcoming of our research. In our study, we used H&E staining to evaluate the effect of GFT505 on NASH inflammation, and did not detect which inflammatory cells decreased after GFT505 treatment. We will detect inflammatory cells in future experiments. Truly thanks for your useful comments.

12. In this study, is GFT505 used as a preventive or a curative treatment?

Reply: in our study, GFT505 was used as curative treatment. 

13. What are the effects of GFT505 in control diet-fed mice?

Reply: thanks for your review. We haven’t test the effects of GFT505 in control diet-fed mice in our study. This is also one of our research shortcomings. We did not consider the side effects or good effects of GFT505 on normal mice. In the future, we will design related experimental groups.

---

## [Decision Letter · Decision Letter 1]

22 Sep 2020

PONE-D-20-10913R1

The pharmacodynamic and differential gene expression analysis of PPAR α/δ agonist GFT505 in CDAHFD-induced NASH model

PLOS ONE

Dear Dr. Yao,

Thank you for submitting your manuscript to PLOS ONE. After careful consideration, we feel that it has merit but does not fully meet PLOS ONE’s publication criteria as it currently stands. Therefore, we invite you to submit a revised version of the manuscript that addresses the points raised during the review process.

What was the normal control diet?

Please add the catalog numbers for the commercially purchased kits (cholesterol, triglyceride, etc...) and antibodies used. Many companies have multiple version of similar products.

I agree with the reviewer that the lack of a control group treated with the GFT505 is a major limitation of the study designed and needs to be addressed in the discussion as a  limitation to the interpretation of the study findings.

I agree with your response to the reviewer comments that histological staining is the standard for fibrosis evaluation, I suggest probing via western blot for an additional marker of fibrosis (or try another collagen antibody), further these data should be included in the manuscript along with the description of the methodology for the tissue processing for western blots. Especially as the alpha-SMA western data showed a reduction.

It would be a nice addition to the study if the RNA-seq data showed changes to the type of inflammatory cells present in the liver, or as the reviewer suggests run RT-PCR for markers such as CD45+ and CD163+ (or histological staining for markers for inflammatory cells), while these experiments are not absolutely necessary I agree with the reviewer that this addition would greatly strengthen the manuscript

We look forward to receiving your revised manuscript.

Kind regards,

Jonathan M Peterson, Ph.D.

Academic Editor

PLOS ONE

Additional Editor Comments (if provided):

What was the normal control diet?

Please add the catalog numbers for the commercially purchased kits (cholesterol, triglyceride, etc...) and antibodies used. Many companies have multiple version of similar products.

I agree with the reviewer that the lack of a control group treated with the GFT505 is a major limitation of the study designed and needs to be addressed in the discussion as a limitation to the interpretation of the study findings.

I agree with your response to the reviewer comments that histological staining is the standard for fibrosis evaluation, I suggest probing via western blot for an additional marker of fibrosis (or try another collagen antibody), further these data should be included in the manuscript along with the description of the methodology for the tissue processing for western blots. Especially as the alpha-SMA western data showed a reduction.

It would be a nice addition to the study if the RNA-seq data showed changes to the type of inflammatory cells present in the liver, or as the reviewer suggests run RT-PCR for markers such as CD45+ and CD163+ (or histological staining for markers for inflammatory cells), while these experiments are not absolutely necessary I agree with the reviewer that this addition would greatly strengthen the manuscript

Reviewers' comments:

Reviewer's Responses to Questions

**Comments to the Author**

1. If the authors have adequately addressed your comments raised in a previous round of review and you feel that this manuscript is now acceptable for publication, you may indicate that here to bypass the “Comments to the Author” section, enter your conflict of interest statement in the “Confidential to Editor” section, and submit your "Accept" recommendation.

Reviewer #2: (No Response)

2. Is the manuscript technically sound, and do the data support the conclusions?

Reviewer #2: Partly

3. Has the statistical analysis been performed appropriately and rigorously? 

Reviewer #2: Yes

4. Have the authors made all data underlying the findings in their manuscript fully available?

Reviewer #2: Yes

5. Is the manuscript presented in an intelligible fashion and written in standard English?

Reviewer #2: Yes

6. Review Comments to the Author

Reviewer #2: I thank the authors for their responses. However, I think they did not address some of, what I think are important, suggestions. For this paper to be scientifically rigorous and sound, these suggestions need to be addressed. Please, find these suggestions below:

1. What are the effects of GFT505 in control diet-fed mice?

2. What types of inflammatory cells are reduced after GFT505 treatment? You can check these by simple qPCR for markers of inflammatory cells.

3. Figure 3 and R-figure 1: it is understandable that for NASH model, collagen WB is difficult to get. However, collagen protein levels are a standard way to check liver fibrosis. Therefore, either the authors could try to get a good collagen antibody or perform hydroxy proline assay. Please include these panels in the main Figure 3.

7. PLOS authors have the option to publish the peer review history of their article (what does this mean?). If published, this will include your full peer review and any attached files.

Reviewer #2: No

---

## [Author Response · Author response to Decision Letter 1]

20 Nov 2020

Editor’s advice:

1. Please add the catalog numbers for the commercially purchased kits (cholesterol, triglyceride, etc...) and antibodies used. Many companies have multiple version of similar products.

Reply: thanks for your advice. We have add the catalog numbers for the commercially purchased kits and antibodies used in our manuscript.

2.I agree with the reviewer that the lack of a control group treated with the GFT505 is a major limitation of the study designed and needs to be addressed in the discussion as a limitation to the interpretation of the study findings.

Reply: Thanks for your suggestion. We haven’t consider the effects of GFT505 in control diet-fed mice in our study, which is also neglected by many studies about GFT505 (1,2,3). This is also one of our research shortcomings and we have addressed this in the discussion as a limitation of the study. In the future, we will design related experimental groups. 

3.I agree with your response to the reviewer comments that histological staining is the standard for fibrosis evaluation, I suggest probing via western blot for an additional marker of fibrosis (or try another collagen antibody), further these data should be included in the manuscript along with the description of the methodology for the tissue processing for western blots. Especially as the alpha-SMA western data showed a reduction.

Reply:Thanks for your advice. We bought a new antibody of collagen 1 (HuaBio, CAT# RT1152). The results of western blot was showed in R-figure 1 and there wasn’t clear bind of collagen 1 in our results. It is difficult for to evaluate the protein level of collagen 1 by western blot in NASH animal model. One of the characteristics of NASH is steatosis and fat has a great influence on the extraction of tissue protein. Actually, collagen 1 evaluation in many studies about NASH is mainly dependent on Immunohistochemistry and Sirus red staining (1,3). Moreover, while the evaluation of collagen 1 or collagen 3 is most dependent on the tertiary/quaternary structures, the Immunohlistochemistry (IHC) is the standard method for collagen evaluation (2). Considering the importance of Collagen1 levels and the concerns of editor and reviewer, we run the RT-qPCR of collagen 1 (R-figure 2) in three groups based on our collagen subtype mRNA expression including col1a1, col1a2, col3a1 as a supplement to the IHC of collagen 1.

4.It would be a nice addition to the study if the RNA-seq data showed changes to the type of inflammatory cells present in the liver, or as the reviewer suggests run RT-PCR for markers such as CD45+ and CD163+ (or histological staining for markers for inflammatory cells), while these experiments are not absolutely necessary I agree with the reviewer that this addition would greatly strengthen the manuscript.

Reply: Thanks for your professional comments. The results of RT-qPCR experiment (R-figure 3) showed that the mRNA expression of CD45 in the Vehicle group was higher than that in the Control group, which was decreased after the treatment of GFT505. The mRNA expression of CD163 in the Vehicle group was lower compared with control group, which was increased after the treatment of GFT505. 

Macrophages are heterogeneous and their phenotype and functions are regulated by the surrounding micro-environment (1). Macrophages commonly exist in two distinct subsets: classically (M1) activated and alternatively (M2) activated (2,3). M1-macrophages are pro-inflammatory and M2-macrophages are anti-inflammatory (1). CD45 is one of the makers of M1-macrophage and CD163 is one of the makers of M2-macrophage. Our results shows that the vehicle group has the highest CD45 mRNA expression and the lowest CD163 mRNA expression, which are corresponding to previous studies(2,3,4).

Reviewers' comments:

1.What are the effects of GFT505 in control diet-fed mice?

Reply: Thanks for your suggestion. We haven’t consider the effects of GFT505 in control diet-fed mice in our study, which is also neglected by many studies about GFT505 (1,2,3). This is also one of our research shortcomings and we have addressed this in the discussion as a limitation of the study. In the future, we will design related experimental groups. 

2.What types of inflammatory cells are reduced after GFT505 treatment? You can check these by simple qPCR for markers of inflammatory cells.

Reply:Thanks for your advice. We evaluated the CD45-represented M1-macrophages and CD163-represented M2-macrophages via RT-qPCR. The results of RT-qPCR experiments (R-figure 3) showed that the mRNA expression of CD45 in the Vehicle group was higher than that in the Control group, which was decreased after the treatment of GFT505. The mRNA expression of CD163 in the Vehicle group was lower compared with control group, which was increased after the treatment of GFT505. Macrophages are heterogeneous and their phenotype and functions are regulated by the surrounding micro-environment (1). Macrophages commonly exist in two distinct subsets: classically (M1) activated and alternatively (M2) activated (2,3). M1-macrophages are pro-inflammatory and M2-macrophages are anti-inflammatory (1). CD45 is one of the makers of M1-macrophage and CD163 is one of the makers of M2-macrophage. Our results shows that the vehicle group has the highest CD45 mRNA expression and the lowest CD163 mRNA expression, which are corresponding to previous studies(2,3,4).

3.Figure 3 and R-figure 1: it is understandable that for NASH model, collagen WB is difficult to get. However, collagen protein levels are a standard way to check liver fibrosis. Therefore, either the authors could try to get a good collagen antibody or perform hydroxyproline assay. Please include these panels in the main Figure 3.

Reply: Thanks for your suggestion. According to your suggestion, we bought a new Hydroxyproline assay kit (Nanjing Jiancheng Bioengineering Institute, A030-2-1) and performed a hydroxyproline assay (R-figure 4). The results showed that there was no difference between vehicle group and GFT505 group. Hydroxyproline is essential to collagen synthesis, thus the level of hydroxyproline may be an indirect index for fibrosis (1). In fact, many studies of NASH didn’t choose hydroxyproline as a fibrosis evaluation index. Moreover, fibrosis evaluation is comprehensive and dependent on multiple indexes. According to the results in Figure 2B, 3D and 5D, GFT505 has a favorable effect in inhibiting fibrosis. We bought a new antibody of collagen 1 (HuaBio, CAT# RT1152). The result of western blot was showed in R-figure 1 and there wasn’t clear bind of collagen 1 in our results. It is difficult for to evaluate the protein level of collagen 1 by western blot in NASH animal model. One of the characteristics of NASH is steatosis and fat has a great influence on the extraction of tissue protein. Actually, collagen 1 evaluation in many studies about NASH is mainly dependent on Immunohistochemistry and Sirus red staining (1,3). Moreover, while the evaluation of collagen 1 or collagen 3 is most dependent on the tertiary/quaternary structures, the Immunohlistochemistry (IHC) is the standard method for collagen evaluation(2). Considering the importance of Collagen1 levels and the concerns of editor and reviewer, we run the RT-qPCR of collagen 1 (R-figure 2) in three groups based on our collagen subtype mRNA expression including col1a1, col1a2, col3a1 as a supplement to the IHC of collagen 1.

---

## [Decision Letter · Decision Letter 2]

1 Dec 2020

The pharmacodynamic and differential gene expression analysis of PPAR α/δ agonist GFT505 in CDAHFD-induced NASH model

PONE-D-20-10913R2

Dear Dr. Yao,

We’re pleased to inform you that your manuscript has been judged scientifically suitable for publication and will be formally accepted for publication once it meets all outstanding technical requirements.

Kind regards,

Jonathan M Peterson, Ph.D.

Academic Editor

PLOS ONE

Additional Editor Comments (optional):

Reviewers' comments:

Reviewer's Responses to Questions

**Comments to the Author**

1. If the authors have adequately addressed your comments raised in a previous round of review and you feel that this manuscript is now acceptable for publication, you may indicate that here to bypass the “Comments to the Author” section, enter your conflict of interest statement in the “Confidential to Editor” section, and submit your "Accept" recommendation.

Reviewer #2: All comments have been addressed

2. Is the manuscript technically sound, and do the data support the conclusions?

Reviewer #2: Partly

3. Has the statistical analysis been performed appropriately and rigorously? 

Reviewer #2: Yes

4. Have the authors made all data underlying the findings in their manuscript fully available?

Reviewer #2: Yes

5. Is the manuscript presented in an intelligible fashion and written in standard English?

Reviewer #2: Yes

6. Review Comments to the Author

Reviewer #2: Thank you for the responses. The authors have responded to all my comments and I have no further suggestion.

7. PLOS authors have the option to publish the peer review history of their article (what does this mean?). If published, this will include your full peer review and any attached files.

Reviewer #2: No

---

## [Editor Report · Acceptance letter]

7 Dec 2020

PONE-D-20-10913R2 

The pharmacodynamic and differential gene expression analysis of PPAR α/δ agonist GFT505 in CDAHFD-induced NASH model 

Dear Dr. Yao:

I'm pleased to inform you that your manuscript has been deemed suitable for publication in PLOS ONE. Congratulations! Your manuscript is now with our production department. 

Kind regards, 

on behalf of

Dr Jonathan M Peterson 

Academic Editor

PLOS ONE